# Formation and characteristics of an Ultisol in Peninsular Malaysia utilized for oil palm production

Arolu Ayanda Fatai<sup>1</sup>, Jusop Shamshuddin<sup>1</sup>, Che Ishaq Fauziah<sup>1</sup>, Othman Radziah<sup>1</sup> and Mohsen Bohluli<sup>1</sup>

<sup>1</sup>Department of Land Management, Faculty of Agriculture, Universiti Putra Malaysia, 43400 Serdang, Selangor, Malaysia Correspondence to: J. Shamshuddin (<u>shamshud@upm.edu.my</u>)

Keywords: Acid soil, Oil palm, Soil fertility, Soil mineralogy, Ultisol

# Abstract

Most of the soils in the upland areas of Peninsular Malaysia are classified as Ultisols. Oil palm production on these soils is usually limited by their inherent low soil productivity. However, the crop is cultivated successfully on most of the soils following right soil management practices. A study was conducted in Bera, Malaysia to determine the characteristics and fertility of an Ultisol cropped to oil palm for many years continuously. In this study, the soil in the plantation was sampled, analyzed and classified. The soil under study was formed under tropical environment with udic moisture regime on fine-grained sedimentary rocks mixed with tuffs of Permian age. Due to very long exposure to the condition of high temperature and high rainfall throughout its development, the soil in the area was reddish in color, clayey, deep and highly weathered. The study found that the clay fraction of the soil was dominated by kaolinite, hematite, goethite and gibbsite; hence, the CEC and basic cations were low. Besides, soil reaction was acidic in nature with soil pH slightly below 5, but the exchangeable Al was more than 1 cmol<sub>c</sub>/kg soil. However, it was believed that these inherent characteristics were not expected to significantly affect the production of oil palm grown on the soil. With proper agronomic practices, the area can be utilized for oil palm production sustainably.

## **1.0 Introduction**

Oil palm is the main agricultural commodity in Malaysia that has helped transform its growing economic development. The country, as the second world largest producer of palm oil after Indonesia, has most of its total planted area under matured palms. This shows how significant palm oil industry is to Malaysia, which is a leading exporter of palm oil, producing 26% of the global trade and 11% of oils and fat production to meet world demand (Malaysia Palm Oil Board, 2015). Presently, more than 5 million ha of land in the country is cultivated to oil palm, producing up to 17.73 million tonnes of palm oil and 2.13 million tonnes of palm kernel oil in a year.

The oil palm industry is rapidly expanding due to the increased demand for oil palm products, which is expected to drive oil palm cultivation to a projected worldwide area of about 38 million hectare by 2050 (Corley, 2009), making palm oil the dominant vegetable oil in the globe (Rajanaidul et al., 2000). This implies a need for a higher degree of efficiency in the production of the commodity to meet its increasing demand at the marketplace.

In Malaysia, one of the major constraints to land use efficiency in oil palm cultivation is the infertile nature of her soils. Due to this reason, research and development on the methods of alleviating the problems related to soil fertility is essential for successful oil palm production. Oil palm in Peninsular Malaysia is mostly cultivated on soils classified as Ultisols (Soil Survey Staff, 2014), which is one of the most common soil Order in the tropics. These soils are highly weathered, having nutrients insufficient for sustainable production of oil palm unless they are fertilized (Shamshuddin et al., 2015).

Due to close proximity to the Equator, tropical regions are subjected to intense rainfall and high temperature throughout the year. Because of this, the cation exchange capacity is low, resulting in the low availability of basic cations that limit oil palm production (Paramananthan, 2003). Additionally, Ultisols are acidic in nature, but oil palm is considered as acid-tolerant (Shamshuddin and Auxtero, 1991). The characteristically low pH of Ultisol is due to the presence of soluble aluminum, which may affect the growth of oil palm.

The soils are known to be dominated by secondary minerals such as kaolinite, gibbsite, goethite and hematite in the clay fraction (Shamshuddin and Ismail, 1995). This mineral assemblage forms the clay fraction that provides negative and positive charge in the soils (Palm et al., 2007). Soil mineralogy has been found to strongly influence soil fertility (Shamshuddin et al., 2015).

For about 100 years, Ultisols in Peninsular Malaysia have been utilized successfully for oil palm cultivation despite its low native fertility following appropriate agronomic management practices, with adequate fertilization and conservation practices (Goh et al., 2008). Oil palm

is known to be acid-tolerant although it can grow better if soil pH is raised to an acceptable level that reduces exchangeable aluminum (Shamshuddin et al., 2015).

A proper soil survey and characterization of an area allocated to oil palm cultivation would provide vital information on its suitability. A study was therefore conducted to determine the formation and physico-chemical properties of an Ultisol cultivated to oil palm in Bera, Peninsular Malaysia. Based on the available data from the study, the soil was evaluated for its suitability for sustainable oil palm cultivation.

## 2.0 Materials and Methods

#### 2.1 The study area and soil sampling

The study was conducted in an oil palm plantation belonging to a farmer in Bera, Pahang (GPS 03.27362 N, 102.58044 E). Mapping of the area under investigation was carried out by conventional soil survey techniques, using soil auger to collect samples at pre-determined spots at short distance from each other. Soil color and texture (using finger) were immediately determined. Selected soil samples were subjected to mechanical analysis in the laboratory at Univesiti Putra Malaysia, Serdang, for the determination of particle-size distribution.

Based on color and texture, it was concluded that the area under investigation comprised only one soil type, evenly distributed throughout the oil palm plantation. Then, a soil pit was dug at an appropriate location in the plantation to study its profile in detail. The pedogenetic horizons in the soil profile were identified and subsequently described, followed by collecting soil samples according to the genetic horizons so identified. The samples were brought to the laboratory in Universiti Putra Malaysia at Serdang in preparation for the detailed soil analyses.

Note that the soil in the area was completely weathered at least up to 2 meters depth due to very long exposure to the tropical conditions prevailing in Peninsular Malaysia. As such, the rock outcrops that formed the soil were not able to be identified during the soil survey. However, the rock type forming the soil in the area was later determined using a geology map produced by Mohd Shafeea et al. (2000), which is shown in Figure 1.