# Peer review of "Formation and characteristics of an Ultisol in Peninsular Malaysia utilized for oil palm production"

_Solid Earth, 2017_

## Referee Comment (RC1) · Anonymous Referee #1 · 5 Sep 2017

Dear Editor,

1. the manuscript have several significant flaws, some are highlighted as below. to start with, is the abstract that does not support the study at the least. 2. the study objectives stated are very confusing. in the title the objectives are stated as, formation and characteristics. in the abstract it is stated the determine the characteristics and fertility. in the introduction, stated as formation and physico-chemical properties. which one which is the focus? 3. study area is not clearly designated. 4. soil sampling not stated in clear manner. how many samples were collected to justify the chemical/physical/mineralogical characteristics? 5. what is meant by conventional soil sam-

pling? 6. the author stated that based on colour and texture, the soil is mainly one series only. how did the author come to such a conclusion using 2 parameters only? 7. scientific method and references are not clearly stated. 8. pg.10, Figure 4. unacceptable. the fundamental of soil description is highly flawed. the use of machinery clearly demarcate the horizon. the measure tape is not properly indicative of the soil depth and etc. Only one (1) soil pit for Ultisols used in this study? 9. the author stated Jempol Series? This is likely based on local soil series. what is the international equivalent (classification) of this series based on USDA Soil Taxonomy? how did the author came to the conclusion of Jempol Series? Comparative data from past study is missing. 10. Based on what soil suitability study was conducted? The representation and explanation was not sufficient and supportive at the least. 11. Grammar and syntax error throughout the manuscript. 12. References format, and please re-check.

Many more flaws are noted in the results and discussion. However, not pertinent in explaining any further than above.

Thus, unfortunately, the paper is recommended to be rejected.

---

## Referee Comment (RC2) · Anonymous Referee #2 · 25 Oct 2017

**Formation and characteristics of an Ultisol in Peninsular Malaysia  for oil palm production**

Arolu Ayanda Fatai[1], Jusop Shamshuddin[1], Che Ishaq Fauziah[1], Othman Radziah[1] and Mohsen Bohluli[1]

[revised manuscript text omitted]

(A=Bera Formation; B= Semantan Formation; C= Sandstone)

[Figure]

**2.2 Particle-size analysis**

Soil particle-size distribution was carried out using the pipette method. The textural classification was based on USDA soil texture triangle of size classes as clay (<2 μm), silt (2-50 μm) and sand as (50-2000 μm) (Soil Survey Staff, 2014).

**2.3 Chemical analyses**

Soil pH was determined in water at a soil to solution ratio of 1:2.5 (Jones jr, 2002). The electrical conductivity (EC) was measured in water at a soil to solution ratio of 1:5. Exchangeable cations were extracted using 100 mL of 1 M ammonium acetate buffered at pH 7. The concentrations of K, Ca, Mg and Na in the solutions were determined by the atomic absorption spectrophotometer (AAS) (Ross and Ketterings 1995; Shamshuddin, 2006). Cation exchange capacity (CEC) was determined by leaching the ammonium ions from the exchange sites with 0.05 M $K_2SO_4$ after the soil was leached for the exchangeable cations extraction and the CEC from the extract was determined using an auto analyzer. Prior to that, the soil was washed with 100 mL of ethanol (95%) to remove excess $NH_4^+$ (Jackson, 2005). Total carbon in soil was determined by Dry Combustion techniques using LECO CR-412 carbon analyzer (LECO, corporation. St. Joseph, USA) by weighing one gram of soil into a tarred ceramic boat. Available phosphorus was analyzed using Bray and Kurtz II method (Landon, 2014). In this method, 2 grams of soil was put in a 50 mL volumetric flask and 14 mL of extracting solution (0.03 N $NH_4F$ and 0.1M HCl). Using the wrist inversion techniques, the mixture was inverted for 45 seconds and the extract was filtered using the Whatman no.42 filter paper.

Exchangeable Al was extracted using 1 M KCl (Elisa et al., 2015) and Al in the extract was measured using an inductively coupled plasma optical emission spectrometry (ICP-OES). Fifty mL of 1 M KCl was added into a 5g soil in a plastic vial. This was agitated for 30 minutes and the supernatant was decanted using the Whatman no. 42 filter paper. Total micronutrients content of the soil was extracted using Aqua Regia method (Gray et al., 2006) and the elements in the solutions were determined using AAS.

[Figure]

**2.4 X-Ray Diffraction analysis**

XRD analysis was performed to determine the clay mineralogical composition. During mechanical analysis, when soil was separated into sand, silt and clay after organic matter in the soil has been removed through the action of hydrogen peroxide. The clay was collected, spread on a slide and run on Philip PW 3040/60 X'pert Pro X-ray diffractometer at room temperature (Shamshuddin, 2011), using CuK-alpha radiation target, operated at 40 kV and 30 mA. The oriented specimens were scanned from 3 to 50° 2θ at 1° per minute.

**3.0 Results and Discussion**

**3.1 The geology of the area**

The geology of the area under investigation is depicted in Figure 1 based on the study of Shafeea et al. (2000) who found that Bera area in Pahang was composed mainly of three rock types: 1) Bera Formation; 2) Semantan Formation, and 3) Berangga Sandstone. The soil of the area under study was formed from rocks of Bera Formation of Middle Permian age.

These rocks of Bera Formation consisted predominantly of clastic rocks, including mudstone, shale, sandstone and siltstone mixed with tuffaceous materials. These rocks were partly exposed and/or subjected to several degrees of metamorphism during the history of their existence since Permian era.

The oil palm plantation where the study was conducted happened to be located in the area dominated by fine-grained materials on the basis of the clayey nature of the soils so formed; most probably they were mudstone and shale mixed with tuffs having high amount of oxides of Fe. Such being the case, the color of the soil was reddish in nature. If the parent materials of the soil were partly composed of coarse-grained rocks, such as clastic rocks and/or sandstone, the amount of sand in the soil would have been much higher than that reported in this study.

**3.2 Morphological properties of the soil**

The area under investigation was an oil plantation (Figure 2) which was used for a field trial for oil palm by another group of researchers and the results of which will be published

[Figure]

elsewhere. The numerals shown in the diagrammatic representation of the experimental plot (Figure 3) were the treatment blocks of the said field trial. It was observed that the palm grew quite well and the yield (in terms of fresh fruit bunches) was above the Malaysian average of about 30 t/ha/year (Figure 2), which was considered high.

[Figure]

Figure 2. Oil palm in the plantation in the study area

The current paper is the result of the investigation mainly on the physico-chemical properties of the soil in the field trial. The other objective of the study was to explain the genesis of the soils formed on fine-grained sedimentary mixed with tuffs in geological time. After a long period of intense weathering under tropical conditions, the soil so formed was dominated by secondary minerals of low fertility status. Yet, the yield of the oil palm grown on the soil was high, comparable to that of the other parts of the country. We intend to find out why and how the oil palm performs so well in spite of the fact that the soil is inherently poor in fertility.

[Figure]

[Figure]

[Figure]

Figure 3. Diagramatic representation of the experimental plots in the area under study in Bera, Pahang

Morphological characterization of the soil profile in the field showed the presence of an argillic horizon (Bt), evidenced by the presence of patchy cutans in the subsoil (Table 1; Figure 4), and was confirmed by the accumulation of clay at depth (Table 2). On checking the soil profile in detail, we found that the boundaries demarcating the genetic horizons were diffused, making it difficult to distinguish between them (Table 1). The topsoil has a strong brown color (7.5YR 4/6), which changed to yellowish red (5YR 6/8) in the subsoil. Lower down the profile, the soil color was red 2.5YR4/6. Reddish soil color was indicative of the presence of hematite ($Fe_2O_3$). The soil structure was sub-angular blocky and root abundance decreased down the soil profile.

The soil was moderately drained, probably due to the presence of too much clay throughout the depth of the soil profile. Had there been more sand present in the soil, the drainage class would have been well-drained. The organic matter content of in topsoil was very high,

(c) Author(s) 2017. CC BY 4.0 License.

[Figure]

[Figure]

probably due to the debris from fallen fronds and/or empty fruit bunches which had been laid out in the inter-row of the oil palm. High organic matter in the topsoil could have resulted in soil structural improvement which, to a certain extent, had enhanced soil fertility.

Table 1: Description of the soil profile

| Horizon | Depth | Description |
|---|---|---|
| Ap | 0-18 cm | Strong brown (7.5YR 4/6); clay; moderate, sub-angular blocky; friable; many coarse roots; clear, smooth boundary; |
| Bt $_1$ | 18-57 cm | Yellowish red (5YR 5/8); clay; moderate, sub-angular blocky; friable; thin, patchy cutans; many course roots; diffuse, smooth boundary; |
| Bt $_2$ | 57-85 cm | Red (2.5YR 4/6); clay; moderate, sub-angular blocky; friable; thin, patchy cutans; few coarse roots; , diffuse, smooth boundary. |
| Bt $_3$ | 85-146 cm | Red (2.5YR 4/6); clay; moderate, sub-angular blocky; friable; thin patchy cutans; few course root; diffuse, smooth boundary. |

[Figure]

[Figure]

[Figure]

Figure 4. The profile of Jempol Series

Table 2: Particle-size distribution of the soil under study

| Horizon | Depth (cm) | Sand (%) | Silt (%) | Clay (%) | Texture |
|---------|-----------|----------|----------|----------|---------|
| $Ap_1$ | 0-18 | 7.34 | 14.88 | 75.78 | Clay |
| $Bt_1$ | 18-57 | 4.03 | 9.39 | 84.58 | Clay |
| $Bt_2$ | 57-85 | 3.87 | 7.63 | 88.05 | Clay |
| $Bt_3$ | 85-146 | 5.08 | 9.49 | 84.43 | Clay |

The soil can be classified as Jempol Series based on the criteria set by the System of Soil Classification in Peninsular Malaysia (Paramananthan, 1987). According to Tessens and Shamshuddin (1983), this soil series was formed on sedimentary and/or metamorphic rocks mixed with tuffs. Further, the researchers classified the soil of Jempol Series found in Pahang some 50 km away from the site of the current study (Bera) as an Ultisol with low fertility.

[Figure]

**3.3 Mineralogy of the clay fraction**

. Kaolinite was found to be the most abundant clay mineral in the studied soil (Figure 5). Goethite and hematite were also found in the soil, explaining its reddish to yellowish coloration. The X-ray diffractograms showed peaks at 7.2 and 3.59 Å (kaolinite), 2.69Å (hematite) and 4.19 Å (goethite) in the Ap horizon, proving the presence of the minerals so mentioned. The same mineralogical assemblage occurs in the genetic horizons of the subsoil. The presence of these minerals  CEC, which  reduced soil productivity . The presence of the mentioned minerals in the Ultisol under study is consistent with that found by the previous studies of Tessens and Shamshuddin (1983) and Shamshuddin and Fauziah (2010). Knowing the mineralogical assemblage of the clay fraction of the soil is crucial in assessing the important physical and chemical properties related to the growth of oil palm.

[Figure]

Figure 5. XRD diffraction patterns of oriented clays from the various horizons of the studied soil

[Figure]

[Figure]

**3.4 Physico-chemical characteristics of the soil**

The soil under study was Jempol series, which was dominated by the clay fraction in all the genetic horizons, having more than 75% clay content (Table 2). As such, the texture of all horizons was clay. Soils, which are clayey in nature, are thought to originate mainly from fine-grained rocks, such as shale, which was consistent with the study of Reineck and Singh (1973).

The occurrence of patchy cutans (Table 1) observed in the subsoil during the field work was indicative of the presence of argillic horizon (Bt). The clay content was the highest in the $Bt_2$ horizon, proving yet again that there was clay accumulation in that zone. The increase in the clay content with depth is thought to be a result of the translocation/illuviation of the clay in the soil during its formation (Hattori et al., 2005). Due to low basic cations together with low CEC, it was confirmed that the soil was an Ultisol as defined by Soil Taxonomy (Soil Survey Staff, 2014).

High clay content in the soil means that water movement down the soil profile is  restricted. Nonetheless, this is not expected to affect the growth of oil palm significantly as the crop can grow even in the swampy areas.

Soil structures of the topsoil were found to be good, probably due to the presence of sufficient amount of organic matter . Hence, oil palm roots can penetrate the soil profile down to 1 meter depth easily, in the process of seeking for water, especially during dry period (Figure 4). This was confirmed by the presence of some coarse roots at the depth of 85 cm and even below this; the rooting depth of oil palm is normally 30 cm (Tinker and Nye, 2000). Oil palm roots were mainly confined to the topsoil (Figure 4). Note that the roots occurring below 40 cm depth were mostly the coarse ones. This means than that oil palm is a surface feeder, getting its nutrients mainly by the fine roots existing in the topsoil.

The soils collected from the four genetic horizons have a slightly acidic reaction in water (Table 3); soil pH obtained by the current study was typical of a Malaysian Ultisol, having some soluble aluminum in the soil solution, thereby causing a decrease in soil pH (Shamsuddin and Fauziah, 2010). Figure 6 shows the relationship between soil pH and exchangeable Al. As the exchangeable Al in the soil increased, soil pH decreased. This phenomenon can be explained by the hydrolysis of $Al^{3+}$ in the soil solution. The reaction can be depicted as follows:

$$Al^{3+}.6H_2O + H_2O = Al^{2+}OH.5H_2O + H_3O$$

[Figure]

Table 3: Chemical properties of Jempol Series

| Horizon | Depth (cm) | pH H$_2$O | EC (dS/m) | Total C (%) | Total N (%) | Available P mg kg$^{-1}$ |
|---------|-----------|-----------|-----------|-------------|-------------|--------------------------|
| Ap | 0-18 | 4.92 | 0.13 | 1.05 | 0.13 | 3.92 |
| Bt1 | 18-57 | 4.62 | 0.07 | 0.77 | 0.12 | 2.30 |
| Bt2 | 57-85 | 4.09 | 0.15 | 0.27 | 0.08 | 2.98 |
| Bt3 | 85-146 | 3.66 | 0.22 | 2.60 | 0.26 | 2.23 |

As the pKa of Al is 5, soil solution pH will go towards the value of 5 to achieve the state of equilibrium. Additionally, if sufficient amount of $Fe^{3+}$ is present, solution pH will then be going towards 3 because the pKa of Fe is 3 (Shamshuddin et al., 2015). In the end, the equilibrium soil solution pH is somewhere between 3 and 5. This is the reason why the pH of highly weathered Peninsular Malaysian soils is about 4.5 on the average (Shamshuddin and Fauziah, 2010). As such, the pH of the soil under study was 3.7-4.9 (Table 3).

Reverse reaction can also occur if soil pH is increased due to soil management practices. This happens when ammonium sulfate is applied continuously onto the soil to supply nitrogen to fulfill the requirement oil palm growth in order to sustain production. This phenomenon occurs due to specific adsorption of $SO_4^{2-}$ onto the oxide of Fe in the soil (Shamshuddin et al., 2015). For the soil under study, it is hematite ($Fe_2O_3$) that could play the role of the Fe oxide so mentioned.

[Figure]

Figure 6. Relationship between soil pH and exchangeable Al

[Figure]

Aluminum in the soil solution of Ultisols in Malaysia is often present at a toxic level for some crops (Shamshuddin et al., 2015). If soil pH is increased to a level above 5, Al will be precipitated as inert Al-hydroxides (as explained by the equation on Al hydrolysis given earlier); thus, no longer causing any problem to the roots of the oil palm growing on the plantation. It is a good agronomic practice to increase soil pH to a level above 5. However, liming to increase soil pH to sustain production is not recommended because it is too costly due to the large area being cropped to oil palm in the country. For the soil under study, soil pH was slightly below 5 in topsoil, with values decreasing with depth (Table 3).

Chemical analyses of soil samples showed that the exchangeable bases were within the acceptable range (Table 3) and likewise the CEC was within the range expected for a typically highly weathered soil in the tropics (Table 4). High organic matter content as reflected by the high nitrogen and carbon content in the topsoil is thought to be due to the felled fronds and empty fruit bunches that were previously applied in the inter-rows of the oil palm. The amount of carbon and nitrogen was sufficient for the growth of oil palm in the plantation as reflected by the high yield obtained. Micronutrient contents (Mn and Zn) in the soil are sufficient for the requirement of oil palm growth to sustain normal production (Table 5).

Table 4: Exchangeable cations and CEC of Jempol Series

| Horizon | Ca | Mg | K | Na | Al | CEC |
|---------|------|------|------|------|------|------|
| | | | $cmol_c kg^{-1}$ | | | |
| Ap | 0.64 | 0.23 | 0.13 | 0.04 | 1.41 | 7.11 |
| Bt1 | 0.44 | 0.15 | 0.19 | 0.04 | 1.65 | 5.7 |
| Bt2 | 0.83 | 0.46 | 0.08 | 0.04 | 1.67 | 5.78 |
| Bt3 | 1.56 | 1.66 | 0.15 | 0.04 | 1.81 | 10.1 |

Table 5: Micronutrient contents in the soil of Jempol Series

| Cu | Fe | Mn | Zn |
|------|--------|-------|------|
| | | mg/kg | |
| 64.5 | 7152.2 | 283.5 | 67.7 |
| 63.4 | 7123.5 | 274.5 | 55.8 |
| 62.1 | 7163.3 | 266.1 | 57.4 |
| 50.5 | 7119.7 | 339.5 | 53.8 |

[Figure]

[Figure]

The available phosphorus in the soil was low (Table 3). However, the low level of available P in the soil was not reflected by the yield of the oil palm in terms of fresh fruit bunches. The oil palm in the plantation under study was reported by the owner to produce among the highest yield in the area. This was, perhaps, due to the proper agronomic management by its owner which had applied fertilizer regularly at the time when it was needed. Low P availability in the soil could be due to immobilization via specific adsorption of the nutrient by Fe and/or Al, forming Fe-P and Al-P compound (Fageria, and Baligar, 2008). Iron content in the soil was found to be high (Table 5), which was reflected by its reddish color. XRD analysis shown in Figure 5 proved the presence of reddish hematite ($Fe_2O_3$) as well as goethite (FeOOH).

**3.5 Soil genesis and classification**

To understand the formation of the soil, its profile description was studied in detail. This was followed by studying the physico-chemical properties of the soil obtained from laboratory analyses. Besides, the geology of the area had to be understood so that rocks forming the soil were identified. We believed that the soil formed from fine-grained sedimentary rocks mixed with tuffs of very old age (Permian). These rocks were subjected to a very long process of weathering under tropical conditions, resulting in the development of a very deep soil. Due to the prevailing high temperature and rainfall, the soil had adequate moisture throughout the year. This condition is referred to as the udic moisture regime (Soil Survey Staff, 2014). In the end, the mineralogy of the soil would be dominated by the secondary minerals, such as kaolinite and the oxides of Fe and Al, which had far reaching consequences on the inherent fertility of the soil.

The topsoil was yellowish red, which becoming redder with depth. This was probably due to the presence of higher amount of hematite in the subsoil compared to that of the subsoil. Hematite is a very common mineral in the highly weathered and well-drained soil of Peninsular Malaysia (Shamshuddin et al., 2004). The presence of this mineral was confirmed by XRD analysis.

We are convinced that the soil had an argillic horizon evidenced by the occurrence of the patchy cutans in the subsoil. This phenomenon always occurs in the subsoil of Ultisols in Peninsular Malaysia (Shamshuddin and Fauziah, 2010). The evidence for the occurrence of the argillic horizon was supported by the significant increase in the clay content in the $Bt_2$ horizon. As such, the subsoil is qualified to be called as an argillic horizon.

[Figure]

The CEC of the soil was rather low. This low value of CEC is associated with the dominant presence low activity clay such as kaolinite. The presence of gibbsite, hematite, and goethite further contributed to the low CEC value. Low basic cations in the soil were related very much to the low CEC. Under strongly leaching environment prevailing in the soil, much of the basic cations were lost to the underground water during rainy season. Using the available data such as diagnostic horizon (Bt), CEC, cations and other, the soil can be classified according to Soil Taxonomy (Soil Survey Staff, 2014). Based on the system, the soil was classified as clayey, isohyperthermic family of Typic Paleudults.

**3.6 The suitability of the soil for oil palm cultivation**

Oil palm can be cultivated on a wide range of soils in the tropics (Hartley, 1988; Piggot, 1990; Sugandi, 2005). Furthermore, the growth, yield, and management of oil palm are greatly influenced by the soil type and climatic conditions of the area where oil palm is being cultivated (Goh and Chew, 1995; Paramananthan, 2000; Turner and Gillbanks, 2003). The agronomic parameters used to select suitable areas for oil palm cultivation are the soil properties related to growth limitations of oil palm (Wong, 2009) and the terrain of the growing areas as well as climatic requirement (Mantel et al., 2007). Matching available soil information with the requirement would produce a rating that indicates the capability of the land under study to support oil palm cultivation.

Oil palm is most suitable and commonly cultivated in humid tropical climatic regions where rain is abundant throughout the entire year (Corley and Tinker, 2003). The optimal yield per hectare of oil palm requires rainfall of 2500 to 3500 mm or more distributed evenly (IPI, 1991); thus, providing the soil with adequate moisture.

Bera, where the oil palm plantation was located, had a mean annual rainfall of 2000-2500 mm/year and a yearly temperature range of 25-30º C. The relative humidity in the area was between 80-90%. According to the Koppen-Greiger system, the area was classified as having a tropical rainforest climate (Af) (Malaysian Meteorological Department, 2016). The soil in the area under study had adequate moisture, having udic regime throughout the year (Hartley, 1988).

The terrain of the oil palm plantation under discussion was gently sloping. According to shamshuddin et al. (2015) this elevation was considered suitable for oil palm cultivation. The

[Figure]

[Figure]

soil was fine textured and so can retain sufficient moisture; thus, the drainage was excellent for oil palm cultivation. The soil was deep, moderately drained and having no stones or laterite in its profile. This indicates that root penetration into the soil is easy and as such oil palm grown on the soil will have good anchorage, high access to soil moisture with a high ability to exploit nutrients. Oil palm roots were even found at the depth of about 85 cm (Figure 4). The structures of the soil was sub-angular blocky; thus, it is physically suitable for cultivating oil palm.

**Table 6** Criteria for assessing the severity of soil limitations for oil palm cultivation

| Soil properties | Desirable range | Minor limitation | Serious limitation | Very serious limitation |
|---|---|---|---|---|
| Terrain (°) | 0-12 | 12-16 | 16-24 | >24 |
| Effective soil Depth (cm) | >90 | 60-90 | 30-60 | <30 |
| Stoniness (%) | 0-5 | 5-20 | 20-40 | >40 |
| Texture | sandy clay loam, clay loam | loam, sandy loam | loamy sand | sand |
| Structure | well developed | moderately developed | very weak, massive | structureless |
| Water table (cm) | 75-90 | 60-75 | 30-60 | <30 |
| Soil pH | >4.0 | 3.5-4.0 | 3.0-3.5 | <3.0 |

Source: Shamshuddin et al. (2015)

It was found that the topsoil pH was slightly below 5 (Table 3). Although the soil was acidic in nature, oil palm should be able to grow well as it is acid-tolerant; it can even survive at the pH of 4.3 (Auxtero and Shamshuddin, 1991). In terms of soil pH, the soil is suitable for oil palm cultivation (Shamshuddin et al., 2015). The possible problem is the lack of nutrients in the soil, especially available P which has to be regularly applied (Mutert, 1999; Shamshuddin and Noordin, 2011). Having the properties mentioned above, we believed that the soil is suitable for oil palm cultivation. However, it needs to be regularly checked so that enough nutrients are present in the soil for sustainable oil palm production.

[Figure]

[Figure]

**4.0 Conclusions**

The soil in Bera, Peninsular Malaysia used for the current study can be classified as an Ultisol, evidenced by the occurrence of Bt diagnostic horizon. It was formed under tropical environment on fine-grained sedimentary rocks mixed with tuffs of Permian age, and due to very long exposure to high temperature and high rainfall, the soil was highly weathered. The mineralogy of the clay fraction of the soil was dominated by kaolinite, hematite, goethite and gibbsite. Because of this, the CEC and eventually basic cations were low. Additionally, soil reaction was acidic, with pH slightly below 5. The above-mentioned properties were not expected to significantly affect the sustainable production of oil palm. The suitability of the area for oil palm cultivation is further evidenced by the presence of udic moisture regime throughout the year. Hence, the soil under study is suitable for oil palm cultivation provided proper agronomic management is practiced.

**ACKNOWLEDGEMENT**

We wish to express our sincere gratitude to Universiti Putra Malaysia and Lynas Corporation for the technical and financial supports during the conduct of the research.

---

## Author Comment (AC1) · 20 Nov 2017

NO. Referees comments Author's response Page No 1 The abstract does not support the study. The abstract has been revised/improved. The title of the paper has been re-phrased. 1. 2 The study objectives are very confusing. The objective has been re-phrased. 3. 3 The study area is not clearly designated. This has been explained in the methodology. Additional information has been added in the text 3. 4 The soil sampling was not clearly stated. Additional information has been added in the text 3. 5 What is meant by conventional soil sampling? This is the method commonly used in Peninsular Malaysia (Paramananthan, 1987), following standard soil survey techniques

used worldwide (for example FAO). Sentences the text have been revised/improved accordingly. 3. 6 The author stated that based on colour and texture, the soil is mainly one series. How did the author come to such a conclusion using two parameters only? We have checked the geology of the study area. The soil is developed from the same parent material. In Peninsular Malaysia, the first criterion to define soil series is parent material. We checked the soil colour, texture and others as required for the identification of soil series (Paramananthan, 1987) in detail throughout the area using a soil augur during the detailed soil survey (the distance between observation points was 100 m). Sometimes we also checked the soil in between. From these observations, we concluded that only one soil series existed in the study area. Therefore, we dug a soil profile at a suitable site in the study area to study the soil in detail – for its mineralogy, chemical properties and classification (Soil Taxonomy). 3 7 Scientific methods and references are not clearly stated. Methods and references have been checked and improved accordingly 8 Page 10, figure 4 is highly unacceptable. The fundamentals of soil description are highly flawed. The use of machinery clearly demarcates the horizon. The measuring tape is not clearly indicative of soil depth.We have problem with drought during the period of the study, particularly when soil pit was dug. There was no rain for a month soon after finishing the soil survey. We hired a contractor to dig up the soil pit. Bera is far from our university. We to travel about 400 km after the soil pit was dug. The soil was dry and hard – like rock. With over 70% clay content, it was like cement. Difficult even to get the soil sample, let alone to smoothen the soil profile during observation. So we decided to let stay as it was. With difficulty, we described the soil profile. The clay skin (cutans) remained intact and so were the roots, fine and coarse. That was the best we could under the circumstances. We determine soil depth using another tape. The one shown in Figure 4 is just for taking a photo of the soil profile. 10 9 The author stated Jempol series Âň. This is likely based on local soil series. What is the International equivalent (classification) of this series based on USDA soil taxonomy? How the author did came to conclusion of Jempol series. Comparative data from past study is missing. Jempol Series is the series as

defined by the System of Soil Classification in Peninsular Malaysia (Paramananthan, 1987). The taxonomic classification of the soil is given later, in page 16. The profile was similar to the profile studied earlier by Tessens and Shamshuddin (1983), located 50 km away 10 16 10 Based on what soil suitability was conducted, the explanation and representation were not supportive at the least The evaluation was based on the criteria given in Table 6, page 17. This table was prepared by Shamshuddin et al (2015). Data presented in the data were obtained from various sources, including those reported by the experts who evaluated soils for oil palm production in Malaysia. 17 11 Grammar and syntax error throughout the manuscript The grammar and syntax have been improved Throughout the paper 12 References format and please recheck. The format and references have been checked and improved accordingly.